A sustainable approach for smallholder farmers: evaluation of plant by-products and industrial waste in wheat cultivation

Oguz Muhammet Cagri m.cagrioguz@gmail.com
Department of Field Crops, Ankara University , Ankara , Turkey
Izzo Luigi Gennaro
Electronic publication date: 2025 Jul 29
Publication date: 2025
Volume: 13
Electronic Location ID: e19775
Received 2025 Mar 25; Accepted 2025 Jun 30
Copyright: ©2025 Oguz
Copyright year: 2025
Copyright holder: Oguz
License: This is an open access article distributed under the terms of the Creative Commons Attribution License, which permits unrestricted use, distribution, reproduction and adaptation in any medium and for any purpose provided that it is properly attributed. For attribution, the original author(s), title, publication source (PeerJ) and either DOI or URL of the article must be cited.
License URL: https://creativecommons.org/licenses/by/4.0/

Keywords: Plant by-products, Bioregulators, Sustainable agriculture, Smallholder farmers, Rainfed agriculture

Funding: The authors received no funding for this work.

==============================
Large amounts of waste and by-products are generated during the extraction of essential oils, leading to disposal challenges. Integrating these by-products into the production systems of smallholder farmers may help reduce yield losses caused by environmental stress factors. This study investigated the potential use of plant waste and by-products from aromatic plants as bioregulators in wheat cultivation. Greenhouse and field experiments were conducted to evaluate the effects of these materials on yield parameters and biochemical markers in wheat. The treatments included: T0 (control), T1 (plant extract), T2 (plant biomass), and T3 (plant extract + plant biomass). In the greenhouse experiment, the T3 treatment advanced the onset of tillering by 5.4 days, stem elongation by 5.6 days, and heading by 12 days compared to the control. In the field experiment, T3 reduced the heading onset by an average of 5.27 days relative to the control. Additionally, the highest yield was recorded in the T3 treatment, reaching 264.96 kg da−1. These findings suggest that by-products from aromatic and medicinal plants can serve as cost-effective, sustainable, and environmentally friendly biostimulants to enhance yield in agricultural production.

Introduction

Smallholder farmers cultivate within a small-scale farming model, typically with a land area of less than one hectare and up to 10 hectares (FAO, 2012). Around 84% of farms are smaller than two hectares in the world. Smallholder farmers significantly impact agriculture by managing 12% of cultivated areas (Lowder, Skoet & Raney, 2016). Moreover, smallholder farmers provide around 30% of the world’s total crop production and meet 30–34% of global food demand (Ricciardi et al., 2018; Ritchie, 2021). The rainfed agriculture model is common in smallholder farmer-cultivate regions. Although the importance of rainfed agriculture varies from the region, it provides the majority of economic income and food in developing countries (Mamassi et al., 2023).

Rainfed agriculture is the primary farming model in 75% of global agricultural land, meeting approximately 60% of food needs (Reddy et al., 2015). On the other hand, rainfed agriculture is widely practised in arid areas where crop yields are often threatened by adverse climatic conditions (Bradford et al., 2017). Despite the adverse conditions, approximately 60% of farmers earn income by producing with the rainfed-based production model (Tume et al., 2020). Rainfed agriculture has immense potential to meet the food demands of the world’s growing population (Rostami et al., 2022).

Therefore, efforts to increase productivity in rainfed areas are crucial in reducing yield losses, supporting smallholder farmers and ensuring the sustainability of agriculture. Commonly, wheat farming is based on rainfed in semi-arid to arid regions (Demirel et al., 2023). Irregular rainfall and drought stress caused by climate change are the main reasons for yield and quality loss in these areas (Langridge & Reynolds, 2021). Plants activate several molecular, biochemical and physiological response mechanisms in response to drought stress (Oguz et al., 2022). Arid and semi-arid climate conditions activate drought stress signal formation in plants and cause oxidative damage in plant cells. However, the self-regulatory capacity of plants cannot fully protect against the effects of severe drought stress. Exogenous treatments are required to protect plants in most cases (Zhang et al., 2024).

The sustainable strategies have focused on the development of environmentally friendly innovative interventions to prevent yield losses and increase drought stress tolerance (Oğuz, Oğuz & Güler, 2023). Essential oils and plant extracts obtained from medicinal and aromatic plants have secondary metabolites and phenolic compounds that can act as antioxidants and bioregulators to effectively prevent stress-induced oxidative damage. Exogenous treatments of these compounds have been investigated for the sustainability of agriculture with environmentally friendly bio-based technologies and given successful results. The hydrodistillation method is a traditional and basic process for the extraction of essential oils, bioactive compounds and secondary metabolites from plant tissues (Dorman et al., 2003). The hydrodistillation by-products (plant extract and biomass) contain water-soluble phenolic and bioactive substances such as antioxidant compounds, secondary metabolites, etc. (Oreopoulou, Tsimogiannis & Oreopoulou, 2019). On the other hand, plant extracts and plant biomass produced after hydrodistillation are generally not used and cause waste problems (Ben Farhat et al., 2014; Tsimogiannis et al., 2016). Evaluating the recycling of production waste is vital for maximizing the sustainable use of hydrodistillation by-products and contributing significantly to environmentally responsible practices in manufacturing (Oreopoulou et al., 2018).

The amount of waste produced globally is projected to almost double by 2050 and nearly triple by 2100 (Millati et al., 2019). Approaches to utilizing wastes and by-products as biostimulants are an effective strategy for waste disposal. Plant by-products and agro-industrial green wastes contain a rich variety of organic substances such as various nutrients, organic acids, amino acids, phenolic compounds, cellulose, etc. (Kuppusamy et al., 2015). These organic substances can increase soil organic matter content and soil water capacity, support plant nutrient absorption and improve crop yield and quality (Ashokkumar et al., 2022).

This study aims to use the hydrodistillation by-products as a bioregulator (Fig. 1). It was hypothesized that the hydrodistillation by-products of medicinal plants would have a positive effect on physiological and biochemical growth parameters in wheat. Hydrodistillation by-product plant extract (PE) is used as a seed priming agent. Solid plant biomass (PB) was applied by mixing into the soil. The first experiment was conducted in the greenhouse to investigate the effects of PE and PB on physiological and biochemical indicators at important development stages such as seedling, tillering, stemming and heading. In the second hypothesis, it was accepted that by-product treatments would increase the yield potential in rainfed wheat agriculture. In this study, the potential of using medicinal plant hydrodistillation by-products as bioregulators in sustainable wheat agriculture was investigated.

Figure 1 Essential oil hydrodistillation process and by-product collection, analysis and treatments.

Materials and Methods

Plant material

The Kavılca wheat used in the experiments were obtained from the local farmers in Kars province. Kavılca wheat is a type of awned wheat. Kavılca wheat is tetraploid (2n=4x=28). Triticum dicoccum is considered as Emmer type wheat (Kan et al., 2015). Kavılca is an ancient wheat species cultivated in Anatolia for centuries (Weiss & Zohary, 2011). This variety has survived to the present day thanks to its resistance to cold and drought (Kishii, 2019). Kavılca is rich in fibre and has high protein content. Rosemary (Rosmarinus officinalis L.) plants were obtained from the field of Ankara University Field Crops Department.

Extraction method and GC-MS analysis of essential oils

The leaves of rosemary plants grown in the field of Ankara University Field Crops Department were collected and cut into one cm pieces, and then the oil extraction process was carried out by hydrodistillation method. Freshly chopped plant material (250 g) was inserted into the extraction vessel of the clevenger apparatus with 2,000 ml of distilled water and extracted for 3 h. The stock essential oils obtained as a result of distillation were stored in amber-colored tubes at 4 °C in the refrigerator (Oğuz, Oğuz & Güler, 2023).

Analysis of essential oils was performed using gas chromatography (GC) coupled to a mass selective detector equipped with an HP-5MS (cross-linked 5%). Essential oil samples were passed through the capillary column (50 m × 0.32 mm × 1.2 µm). Helium gas was used as carrier gas. The flow rate is set at 10 psi. The column temperature was 60 °C. It was set to hold at 60 °C for 5 min. Then increase by 2 °C/min to 220 °C, hold it for 20 min (Oğuz, Oğuz & Güler, 2023). The GC-MS solution software version 4.45 (Shimadzu, Kyoto, Japan) was used to acquire data. Bidimensional visualisation was generated using Chromsquare software version 2.3. The PubChem and NIST Chemistry WebBook databases were utilised for the characterization of components of rosemary essential oil (Mwithiga et al., 2022).

Preparation and hormone content analyses of by-products

After the hydrodistillation process, solid plant biomass (PB) and liquid plant extract (PE) were collected. By-products were prepared in different ways for both treatments and hormone content analysis. The PB was ground into powder after drying. PB in powder form was stored at +4 degrees for hormone analysis and treatments. PE was filtered from solid particles and then passed through a 0.22 µm sterile hydrophilic filter (ISOLAB) for the second filtration. Then it was centrifuged at 1,800 RPM for 10 min. The supernatant was stored in amber tubes at 4 °C in the refrigerator for PE treatment and hormone. The process of by-product collection and preparation is illustrated in Fig. 1.

The rosemary hidrodestilation by-products hormone content analyses performing according to Oguz (2024) with small changes. The PB was dissected and crushed in liquid nitrogen. One g of plant tissue was placed in 10 ml of sterile distilled water (100 mg ml−1) in a 50 ml falcon tube and shaken for 24 h on a rotary shaker (200 rpm) at 24 ± 1 °C. Then, the sample was shaken (180 rpm) with methanol (20%) for 10 min at 24 ± 1 °C. Then, it was rinsed with sterile distilled water and centrifuged for 20 min at 10,000 rpm. The supernatant was collected. The PE sample was shaken (180 rpm) with methanol (20%) for 10 min at 24 ± 1 °C and it was rinsed with sterile distilled water and centrifuged for 20 min at 10,000 rpm. The supernatant was collected for hormone content analyses. PE and PB samples were stored in amber tubes in the refrigerator at 4 °C until hormone analyses.

ABA, GA and IAA hormone contents in plant extract solution samples were analyzed by high-performance liquid chromatography (HPLC) analysis using a UV detector and C-18 column (Agilent 1100; Agilent Germany, Waldbronn, Germany). For the identification of hormones, samples filtered through 0.45 millipore filters were injected into the column. Methanol (30%) and aqueous acetic acid (1%) were used as mobile phases. Standardized stock solutions of hormone samples were used to identify and quantify plant hormones (Guo et al., 2022). Each sample was measured three times and the average was calculated as µg ml−1.

Greenhouse experiment and by-products treatments

Surface sterilization of wheat seed was done with NaClO (sodium hypochlorite) before PE seed priming. The sterilization method was modified from Oğuz, Oğuz & Güler (2023). The wheat seeds were washed in 10% NaClO for 10 min with a magnetic stirrer, and then rinsed in sterile distilled water three times for 5 min. Seed priming was done in a plastic bag under dark conditions for 12 h with 10 ml of stock PE for every thousand sterilized seeds. It was planted in 30 × 25 cm plastic bags containing three kg of soil, peat and perlite (1:1:1) mixture. In PB treatments, 100 gr PB was added to each pot. Soil moisture was measured with a soil moisture meter throughout the production period. Greenhouse air temperature (28 °C ±2) and humidity (50% ±5) were kept in balance with the automation control system. Plant pots in the greenhouse were placed in four rows with a distance between rows of 25 cm. The height of the trays where the plant bags are placed from the ground is 90 cm. The ceiling height of the greenhouse is four m. The greenhouse experiments were carried out in 2022–2023.

Greenhouse experiments were carried out according to a randomized plot design with 20 replicates representing T0: control, T1: PE, T2: PB and T3: PE+PB treatment. The analyses were performed on five randomly selected plants representing treatments.

Physiological and biochemical measurements in greenhouse experiment

Tiller numbers (TN) and grain number per spike (GNS) were determined by counting. Thousand-grain weight (TGW) (g) was obtained by weighing with a precision scale. Spike height (SH) (cm) was determined by measuring with a millimetric ruler. Tillering beginning day (TBD) and heading beginning day (HD) were given as the number of days from germination/emergence. Additionally, plant height (cm), chlorophyll content (SPAD-502Plus; KONICA Minolta, Tokyo, Japan), proline, malondialdehit (MDA), and relative water content (RWC) (%) values were measured in four important growth periods as S1: seedling, S2: tillering, S3: stem elongation and S4: heading.

Relative water content (RWC) was determined by drying the leaf sample at 105 °C for 30 min and then at 70 °C until a constant weight was reached. After cooling to room temperature, the samples were weighed and the leaf water content was calculated according to the specified formula RWC (%) = [(FW − DW)/FW] ×100 (FW, fresh weight; DW, dry weight) (Deef, 2007; Oğuz, Oğuz & Güler, 2023).

The proline content of leaf tissues was determined spectrophotometrically according to the method determined by Bates, Waldren & Teare (1973). A total of five g of leaf tissue was treated with 3% sulfosalicylic acid. A total of two ml of the obtained plant extract was taken and two ml of acid ninhidrin and two ml of glacial acetic acid were added to it, and it was kept in a 100 °C water bath for 1 h. Then, four ml of toluene was added to each of the samples in the tubes, which were kept in ice for 5 min, and measurements were made at 520 nm in the spectrophotometer.

The amount of MDA was determined from the methods of Heath & Packer (1968) and Sairam & Saxena (2001) with minor modifications. 0.5 g of leaf sample taken from the plant was homogenized with 10 ml of 0.1% trichloroacetic acid (TCA) and then the homogenate was centrifuged at 15,000 rpm for 5 min. One ml was taken from the supernatant of the centrifuged sample and 0.5% thiobarbituric acid (TBA) dissolved in four ml of 20% TCA was added. The mixture was kept in a 95 °C water bath for 30 min, then quickly cooled in an ice bath and centrifuged at 10,000 rpm for 10 min. After that, the absorbance of the supernatant at 532 and 600 nm wavelengths was determined and the malondialdehyde (MDA) content was calculated.

Field experiment and the experimental design

The field soil was ploughed twice with a plow at a depth of 20–25 cm before sowing. Field experiment was established according to the randomized block design. For each treatment, the experiment was carried out in three replicates. Each plot consisted of 10 rows of two m in length, with a row spacing of 25 cm and 500 seeds per m2 for each trial plot. Field experiment was carried out with T0: control, T1: PE, T2: PB, and T3: PE+PB treatment. Wheat seeds were sterilized before priming. Seed priming was done in a plastic bag under dark conditions for 12 h with 10 ml of stock PE for every thousand sterilized seeds. In the T0 control treatment, seed priming was done with an equal amount of sterile distilled water. 100 g of PB was added to the seed bed together with planting in PB treatment. Sowing was carried out in the second week of October 2022.

To ensure sustainability, any organic/synthetic fertilizers or pesticides were not applied to the plants before or after planting. Weeds in the plot areas were cleaned with hand tools. Rainfed agriculture production was carried out without irrigation, including during planting. The parameter related to the yield examined in the field experiment was carried out on 15 randomly selected plants. Field experiment was carried out in the field of Ankara University Field Crops Department. The research site is located at an altitude of 870 m above sea level, between 39°57′ north latitude and 32°51′ east longitude. The experiment was carried out during the 2022–2023 production seasons. Temperature and precipitation data of the field experiment area for 2022–2024 are given in the Fig. S1.

Soil physico-chemical properties analysis

Soil samples were collected from 0–30 cm depth from three different areas representing the land from which soil was taken in the greenhouse and field experiments. Physico-chemical properties of the soil were determined by the hydrometer method described by Klute & Page (1986). Soil pH and electrical conductivity (EC) were specified by the electrometric method. Available phosphorus (P) was evaluated by the Nelson method (Nelson & Sommers, 1982); total soil organic carbon (TOC) content was determined by the potassium dichromate oxidation method. Nitrogen content (total nitrogen) was determined by the Kjeldahl method (Nelson & Sommers, 1980).

Physiological and biochemical measurements in field experiment

The investigated characteristics are the seedling period plant height (SPH), stem elongation perion plant height (SEPH), heading period plant height (HPH), spike height (SH), flag leaf length (FL), harvest period plant height (HVPH), plant emergence day (ED), Tillering beginning day (TD), stem elongation beginning day (SED), heading beginning day (HD), number of tillers (TN), chlorophyll (spad), number of grains per spike (GNS), thousand grain weight (TGW), grain protein content (GPC), grain nitrogen content (GNC), plant number in m2 (PNM) and yield (kg da−1) were determined.

Grain protein and N analysis

Grain nitrogen and protein analyses were carried out using the Kjeldahl Method using the Buchi Auto Kjeldahl Unit K-370 machine. Wheat grain samples were crushed, ground, and homogenized. One gram of the sample was placed in a Kjehdahl flask. 25 mL of concentrated H2SO4 was added, and the Kjehdahl flask was placed in the combustion set. The temperature was adjusted to increase gradually and the combustion process was carried out up to 400 °C (Büchi K-437 digestion system). After the combustion process, 100 ml of 40% NaOH was added to the Kjehdahl flasks. 50 ml of boric acid solution and 10 drops of indicator solution were added to the Erlenmeyer flask and placed in the distillation device. Distillation was carried out until approximately 100–150 mL of distillate was obtained. As a final step, titration was performed to determine the amount of ammonia retained in the boric acid solution. According to the results obtained, grain nitrogen and protein values were determined.

Statistical analysis

The samples in each group were collected randomly and independently. The data obtained for all the results that were measured and observed in the experiment were subjected to analysis of variance (ANOVA) in the IBM SPSS Statistics 22.0 program. The field and greenhouse experiment used a one factorial completely randomized design to compare treatments (one-way analysis of variance). The data in each group are normally distributed, and the variance of the data in each group is homogeneous. The differences between the means in greenhouse and field experiment were determined using the Duncan test at 0.01 levels (Snedecor & Cochran, 1967).

Results

Soil physico-chemical properties analysis

According to the analysis results, the soil nitrogen (0.14%–0.19%) and organic matter content (0.87%–1.32%) are insufficient. Organic carbon content is also low (0.41%–1.23%). Phosphorus (P2O5) (7.96–8.72 kg da−1) and potassium (K2O) contents (104.9–115.1 kg da−1) are at a moderate level. Salt content is at a harmless level for plants (0.15%–0.19%). The soil has alkaline properties according to its pH value (7.15–8.18). The soil EC (Electrical Conductivity) value is at a moderate level (1.4–1.9 dS/m). This value shows that the soil is slightly salty.

Essential oil content

GC-MS analysis results of the rosemary essential oil are given in Table 1. A high level of 1.8-cineole (31.7%) was determined in the essential oil content. Camphor (14.91%), α-pinene (8.76%), α-terpineol (8.31), borneol (7.01) and camphene (5.89) are other metabolites with high content determined in rosemary essential oil (Table 1).

Table 1 GS-MS analyses result of Rosmarinus officinalis L.

Components	(%)	
α-pinene	8.76	
camphene	5.89	
β -pinene	3.09	
myrcene	0.62	
α-terpinene	1.26	
limonene	2.32	
1.8-cineole	31.7	
γ-terpinene	1.12	
p-cymene	2.69	
terpinolene	0.91	
camphor	14.91	
linalool	2.31	
caryophyllene	1.02	
terpinenol	2.13	
ısoborneol	0.51	
α-terpineol	8.31	
borneol	7.01	

Hormon content of PE and PB

PE and PB hormone contents are given in Fig. 2. PE ABA content was determined as 4.96 µg l−1, and PB ABA content was determined as 4.73 µg l−1. The difference between them is statistically insignificant. GA3 content was determined as 9.49 µg l−1 in PE and 8.99 µg l−1 in PB. The difference between PE and PB is insignificant. IAA content was determined as 11.4 µg l−1 in PE. PB IAA content is 9.47 µg l−1. The difference between PE and PB regarding IAA content is statistically significant (p < 0.01).

Figure 2 By-product hormone content analyses results.

Greenhouse experiment

Plant height, chlorophyll, proline, MDA and RWC content values in four different development periods of T0 (control), T1, T2 and T3 treatments in the greenhouse are shown in Fig. 3. The best plant height during the seedling period was determined in T1 and T3 treatments (14.66 cm and 15.26 cm, respectively) compared to T0 and T2 treatments (Fig. 3A). During the tillering period, T3 and T2 (19.56 and 18.04 cm, respectively) were the treatments in which the best plant height was measured. However, the difference between T2 and T1 treatments was statistically insignificant. During the tillering period, the best plant height was measured in T3 and T2 treatments (19.56 and 18.04 cm, respectively). However, the difference between T2 andT1 treatments was statistically insignificant. The minimum plant height was measured as 34.02 cm in the T3 treatment. Similarly, the difference between T0, T2 and T3 treatments is also insignificant. The difference between the effects of the treatments on plant height at the heading stage is statistically insignificant (Fig. 3A). The image of the plants during the ripening period in the greenhouse experiment is given in Fig. 4.

Figure 3 Plant height, chlorophyll, proline, MDA and RWC content values in four different development periods of T0, T1, T2 and T3 treatments in the greenhouse experiments.

Figure 4 Plants at ripening stage in greenhouse experiment.

T0 control treatment (A), T1: PE treatment (B), T2: PB treatment (C), T3: PE+PB treatment (D) Bar: 10 cm.

The effects of treatments on chlorophyll values in different growth periods are given in Fig. 3B. Chlorophyll content (SPAD) in the seedling period was the highest in T1 and T3 treatments (45.22 and 45.63). Although T0 and T2 values are low, the difference between treatments is statistically insignificant (42.44 and 42.22). The highest chlorophyll value during the tillering period was determined in T3 and T2 treatments (58.08 and 56.30). The lowest chlorophyll value was determined in the T0 treatment. The highest chlorophyll value during the stem elongation period was determined in the T3 treatment. The highest chlorophyll value in the heading period was determined in T2 and T3 treatments and the difference between them was insignificant (63.26 and 65.57, respectively) (Fig. 3B).

Proline analysis results are given in Fig. 3C. The highest proline amount during the seedling period was determined in the T3 treatment (7.81). T0 and T1 treatments had the lowest proline levels (3.28 and 3.70, respectively) and the difference between them was insignificant. During the tillering period, the highest proline amount was determined in T0, T1 and T3 treatments (9.22, 10.32 and 8.79, respectively). The lowest proline amount was determined in the T2 treatment during the tillering period (7.60). The difference between the treatments’ effects on the proline amount during the stem elongation period is statistically insignificant (p > 0.01). The lowest proline values in the heading period were 14.31 and 13.18 in T2 and T3 treatments, respectively (Fig. 3C). The MDA analysis results of the treatments in different growth periods are given in Fig. 3D. The highest MDA value during the seedling period was determined in T1 and T3 treatments (16.29 and 18.46, respectively). The difference between T1 and T3 is insignificant. The highest MDA value during the tillering period was determined as 18.77 in the T2 treatment and 16.85 in the T3 treatment. The difference between T0 and T1 treatment is statistically insignificant. The difference between the effects of the treatments in the MDA analyses during the bolting period is statistically insignificant. Similarly, the effects of the treatments in the analyses at the heading period are insignificant (p > 0.01) (Fig. 3D).

Relative water content values are given in Fig. 3E. The difference between the effects of the treatments on the RWC content is insignificant at the seedling period (p > 0.01). Although the lowest RWC content at the tillering period was determined in the T0 treatment (70.43%), the difference between T0 and T3 (72.22%) was statistically insignificant. The highest RWC content was determined as 74.40% and 74.80% in T1 and T2 treatments, respectively. The highest RWC content at the stem elongation period was determined as 79.43% and 80.36% in T2 and T3 treatments, respectively. However, the difference between T2, T3 and T0 (77.45) is insignificant (p > 0.01). In addition, the difference between T0 andT1 is insignificant (p > 0.01) (Fig. 3). The lowest RWC content at the heading period was determined as 76.40% and 77.56% in T0 and T1 treatments, respectively. The difference between T0 and T1 is insignificant. The highest RWC content was determined as 80.25% and 81.70% in T2 and T3, respectively. Similary, the difference between T2 and T3 is insignificant (p < 0.01) (Fig. 3E).

The results of number of tillers (TN), tillering beginning day (TD), stem elongation beginning day (SED), heading beginning day (HD), flag leaf length (FL) measurements in greenhouse experiments are given in Table 2. The difference between the effects of the treatments on TN was found to be statistically insignificant (p > 0.01). On the other hand, the highest TD was determined in the T0 control (67 days). TD was determined as 64.20 and 64.80 days in T1 and T2 treatments, respectively. The lowest TD number was detected in the T3treatment with 61.60 days (Table 2). The lowest number of SED was determined in the T3 treatment with 82.20. In the T2 treatment, SED was determined as 84.00 days. The difference between T2 and T3 is statistically significant (p < 0.01). Besides, there is no difference in terms of SGS between T0 and T1 treatments (p > 0.01) (Table 2). The highest number of HD was determined in T0 andT1 treatments (181.60 and 181.00, respectively), and the difference between them was statistically insignificant (p > 0.01). HD was determined as 174.40 in the T2 treatment and 169.60 days in the T3 treatment. The difference between T2 and T3 is significant (p < 0.01) (Table 2). The effects of the treatments on FL are given in Table 2. The highest FL was determined as 15.83 cm in the T3 treatment (p < 0.01). The difference between the T2 and T3 is significant. The difference between FL values in T0 and T1 treatment is insignificant (12.54 and 12.75 cm, respectively) (p > 0.01) (Table 2).

Table 2 The analysis and measurements in greenhouse experiments.

	TN	TD	SED	HD	FL	
T0	2.40 ± 0.48a	67.00 ± 0.89a	87.80 ± 0.74a	181.60 ± 1.01a	12.54 ± 0.60c	
T1	2.40 ± 0.48a	64.20 ± 0.74b	87.00 ± 0.63a	181.00 ± 0.89a	12.75 ± 0.63c	
T2	2.60 ± 0.48a	64.80 ± 1.32b	84.00 ± 1.09b	174.40 ± 1.2b	14.53 ± 0.60b	
T3	3.00 ± 0a	61.60 ± 1.01c	82.20 ± 0.74c	169.60 ± 1.2c	15.83 ± 0.57a	
Notes.

TN, Number of tillers; TD, Tillering beginning day; SED, Stem elongation beginning day; HD, Heading beginning day; FL, Flag leaf length (cm).

The difference between the means shown with different letters is significant at the 0.01 level.

The results of spike height (SH), grains number in spike (GNS), thousand grain weight (TGW), grain protein content (GPC) and grain nitrogen content (GNC) are given in Table 3. The highest SH value was determined as 8.04 cm in the T3 treatment (p < 0.01). The difference between the T0, T1 and T2 treatments is statistically insignificant (6.64 cm, 7.10 cm and 7.14 m, respectively) (Table 3). The highest GNS value was determined as 22.00 units in the T3 treatment (p < 0.01). The difference between the GNS values of T0, T1 and T2 treatments (16.60, 17.20 and 17.60 respectively) is statistically insignificant (p > 0.01) (Table 3). TGW values are given in Table 3. The highest TGW was determined as 34.96 g in the T3 treatment. The difference between the determined TGW values of the T0, T1 and T2 treatments (31.38, 31.15 and 32.46 g, respectively) is statistically insignificant (p > 0.01) (Table 3). The GPC and GNC analysis results are given in Table 3. The highest GPC value was determined as 13.64 in the T3 treatment. The difference between the other treatments is statistically insignificant. Similarly, the highest GNC was determined as 1.98 in the T3 treatment. The difference between T0, T1 and T2 is statistically insignificant. In addition, the difference between T2 and T3 is also insignificant (p > 0.01) (Table 3).

Table 3 The analysis and measurements in greenhouse experiments.

	SH	GNS	TGW	GPC	GNC	
T0	6.64 ± 0.32b	16.60 ± 1.35b	31.38 ± 1.30b	11.26 ± 0.66b	1.81 ± 0.04b	
T1	7.10 ± 0.22b	17.20 ± 0.74b	31.15 ± 1.13b	12.19 ± 0.48b	1.83 ± 0.05b	
T2	7.14 ± 0.18b	17.60 ± 1.01b	32.46 ± 1.29b	11.79 ± 0.57b	1.86 ± 0.04ab	
T3	8.04 ± 0.26a	22.00 ± 1.41a	34.96 ± 1.05a	13.64 ± 0.56a	1.98 ± 0.08a	
Notes.

SH, Spike height (cm); GNS, Grains number in spike; TGW, thousand grain weight (g); GPC, Grain protein content; GNC, Grain nitrogen content.

The difference between the means shown with different letters is significant at the 0.01 level.

Field experiment

Seeding period plant height (SPH), stem elongation period plant height (SEPH), heading period plant height (HPH), spike height (SH), flag leaf length (FL), and harvest period plant height (HVPH) results from the field experiment is given in Table 4. The highest SPH were measured as 17.56 cm in the T3 treatment. The difference with the T1 treatment (17.21 cm) was insignificant. The difference between the T2 and T1 treatments is insignificant (16.48 and 17.21 cm, respectively). The minimum SPH was determined as 14.71 cm at T0 treatment (Table 4). The highest SEPH was determined in the T1 treatment as 40.41 cm. The lowest SEPH was measured as 35.10 cm in the T3 treatment (Table 4). The highest value of heading plant height was determined as 66.83 in the T1 treatment. There is no statistical difference between other treatment (p > 0.01). The highest SH was determined as 8.11 cm in the T3 treatment. The difference between T3 and T0 (7.50 cm) is significant. The difference between T3 and T0 (7.50 cm) is significant. The T2 and T3 treatments gave the highest results on FL (16.95 and 16.97, respectively). The difference between T2 and T3 is insignificant. The minimum FL was measured as 14.32 cm in the T0 treatment. HVPH was measured as 81.91 cm in the T1 treatment. The difference between T0 and T2 is insignificant (79.43 and 77.58 cm, respectively). The lowest harvest height was determined in the T3 treatment with a plant length of 76.90 cm (Table 4).

Table 4 Measurement results of the field experiment.

	SPH	SEPH	HPH	SH	FL	HVPH	
T0	14.71 ± 0.78c	37.12 ± 1.88b	64.91 ± 1.66b	7.50 ± 0.58b	14.32 ± 0.91c	79.43 ± 1.45b	
T1	17.21 ± 0.66ab	40.41 ± 1.28a	66.83 ± 1.53a	7.00 ± 0.31c	15.69 ± 0.69b	81.91 ± 1.35a	
T2	16.48 ± 0.74b	36.68 ± 1.10bc	65.02 ± 1.47b	7.34b ± 0.42c	16.95 ± 0.65a	77.58 ± 2.33bc	
T3	17.56 ± 0.73a	35.10 ± 1.96c	63.96 ± 1.53b	8.11 ± 0.44a	16.97 ± 0.82a	76.90 ± 2.22c	
Notes.

SPH, Seedling period plant height (cm); SEPH, Stem elongation period plant height (cm); HPH, Heading period plant height; SH, Spike height (cm); FL, Flag leaf length (cm); HVPH, Harvest period plant height (cm).

The difference between the means shown with different letters is significant at the 0.01 level.

Plant emergence day (ED), Tillering beginning day (TD), Stem elongation beginning day (SED), heading beginning day (HD), number of tillers (TN) and chlorophyll (SPAD) values are given in Table 5. ED was determined as 8.40 and 8.20 days in T0 and T2 treatments, respectively. It was determined as 6.46 and 7.00 days in T1 and T3 treatments, respectively (Table 5). TD was determined as 77.13 in T0 and T1 treatments. In T2 and T3 treatments, TD was determined as 74.40 and 74.13 days, respectively (Table 5). The lowest SED was determined as 91.13 and 90.20 days in T0 and T3 treatments, respectively. In the T1 and T2 treatments, the highest SED were measured as 92.46 and 92.80 days. The difference between T1 and T2 is statistically insignificant. The highest number of days to heading (HD) was determined as 190.73 and 190.93 days in T0 and T1 treatments (Table 5). The difference between treatments T0 and T1 is insignificant. In T2 and T3 treatments, the lowest HD was determined as 189.13 and 185.46 days. The difference between T2 and T3 treatments is insignificant in HD. The TN was determined as 5.13 in the T3 and 4.73 in the T2 treatment. The lowest TN was determined in the T0 treatment (3.73). The number of tillers was determined as 4.00 in the T1 treatment. The highest Chlorophyll (SPAD) amount was measured as 65.73 in the T3 treatment. The difference between the T3 and T1 (61.31) is insignificant. The difference between the chlorophyll content determined in T0 and T2 treatments (55.06 and 59.13, respectively) is statistically insignificant (Table 5).

Table 5 Effect of treatments on development days, tiller number and chlorophyll content in field experiment.

	ED	TD	SED	HD	TN	SPAD	
T0	8.40 ± 0.48a	77.13 ± 1.54a	91.13 ± 1.30b	190.73 ± 1.76a	3.73 ± 0.77c	55.06 ± 4.22c	
T1	6.46 ± 0.80b	77.13 ± 1.30a	92.46 ± 1.14a	190.93 ± 1.61a	4.00 ± 0.73bc	61.31 ± 5.99ab	
T2	8.20 ± 0.74a	74.40 ± 1.08b	92.80 ± 1.46a	189.13 ± 1.54b	4.73 ± 0.77ab	59.13 ± 4.36bc	
T3	7.00 ± 0.73b	74.13 ± 1.14b	90.20 ± 1.27b	185.46 ± 1.14c	5.13 ± 0.71a	65.73 ± 2.61a	
Notes.

ED, Plant emergence day; TD, Tillering beginning day; SED, Stem elongation beginning day; HD, Heading beginning day; TN, Number of tillers.

The difference between the means shown with different letters is significant at the 0.01 level.

Grains number in spike (GNS), thousand grain weight (TGW), grain protein content (GPC), grain nitrogen content (GNC), plant number in per m2 (PNM) and yield (kg da−1) for the treatments in field experiment are given in Table 6. Compared to the T0 treatment, the highest TGW was determined as 25.20 and 23.80 in the T3 and T2 treatments, respectively. The difference between them is statistically insignificant (Table 6). Although the T0 and T1 treatments are 21.66 and 22.33 respectively, the difference between them is insignificant. Similarly, the highest values in TGW compared to the control were determined as 36.36 and 37.35 g in T2 and T3 treatments, respectively. The lowest value was determined as 32.77 g in the T0 treatment, while 34.31 g TGW was determined in the T1 treatment. The difference between T0 and T1 was found to be significant (p < 0.01) (Table 6). GPC analysis was determined as the highest values in T0, T2 and T3 treatments and the difference between them was insignificant (11.84, 11.99 and 12.27, respectively). The difference between the GPC determined in the T1 treatment and the other treatments is significant (Table 6). The highest grain nitrogen content was measured in the T3 treatment (1.98). The difference with other treatments is significant (p < 0.01). The difference between the grain nitrogen values (1.74, 1.76 and 1.83) in the T0, T1 and T2 treatments is insignificant (Table 6) (p > 0.01).

Table 6 Effects of treatments on physiological measurements and yield in field experiment.

	GNS	TGW (g)	GPC	GNC	PNM	Yield ( kg da −1 )	
T0	21.66 ± 1.53b	32.77 ± 1.81c	11.84 ± 0.65a	1.74 ± 0.07b	418.66 ± 4.78d	228.86 ± 3.03c	
T1	22.33 ± 1.39b	34.31 ± 1.34b	11.08 ± 0.48b	1.76 ± 0.12b	438.00 ± 4.32c	235.53 ± 3.31c	
T2	23.80 ± 1.86ab	36.36 ± 0.89a	11.99 ± 0.71a	1.83 ± 0.07b	446.33 ± 5.25b	249.20 ± 2.29b	
T3	25.20 ± 1.97a	37.35 ± 1.18a	12.27 ± 0.44a	1.98 ± 0.09a	457.33 ± 3.85a	264.96 ± 2.81a	
Notes.

GNS, Grains number in spike; TGW, Thousand grain weight (g); GPC, Grain protein content; GNC, Grain nitrogen content; PNM, plant number in per square meter.

The difference between the means shown with different letters is significant at the 0.01 level.

The lowest number of m2 plants was found in the control T0 treatment (418.66). The highest number of m2 plants was determined as 457.33 in the T3 treatment. After T3, the highest m2 plant number value was determined in the T2 application (446.33 plants). More plants were determined per m2 in the T1 treatment compared to the control (438.00 plants). The differences between all treatments are statistically significant. The highest yield (kg da−1) value was determined as 264.96 kg da−1 in the T3 treatment compared to the control T0. The second-better yield value compared to the T0 control was determined as 249.20 kg da−1 in the T2 treatment. The difference between T0 control and T1 treatment is statistically insignificant (Table 6).

Discussion

Türkiye is mainland to many genetic resources that are the ancestors of wheat varieties and are still produced today. Kavılca is an important genetic resource as one of the oldest cultivated and hulled wheat species (Arzani & Ashraf , 2017). Kavılca wheat has high adaptation to difficult climatic conditions both geographically and ecologically (Kan et al., 2015). It is an attractive choice for production by smallholder farmers with these features. Kavılca is cultivated by smallholder farmers in Turkey based on rainfed. For this reason, Kavılca wheat was used as plant material in the study to reflect the production model of smallholder farmers.

In almost every case, smallholder farmers are disproportionately affected by the extreme weather events, rising temperatures and depleting groundwater that are a result of our worsening climate (World Economic Forum, 2022). On the other hand, there is a significant relationship between water scarcity and the nutritional and economic development of smallholder farmers due to their dependence on rainfed agriculture (Vanschoenwinkel & Van Passel, 2018). The use of biostimulants can play an important role in reducing yield losses by improving plant adaptation and plant resistance to water deficit. Biostimulants can also reduce the economic impact of agricultural activities on farmers by reducing the use of fertilizers and pesticides while minimizing the negative impact on the environment (Claros Cuadrado et al., 2019). Organic wastes are generally a by-product generated during different agricultural and industrial production processes such as farms, livestock facilities or food processing plants and are a low-cost resource (Chavan et al., 2022; Zhang et al., 2024). Organic wastes generally contain a rich variety of organic substances such as various nutrients, organic acids, amino acids, cellulose, etc. These organic substances can increase soil organic matter content, promote plant nutrient absorption and hence increase crop yield and quality (Ashokkumar et al., 2022). After physicochemical, biological and phytotoxicity analyses, it was stated that the by-products obtained from essential oil production can be easily used in agriculture (Zaccardelli et al., 2021). The use of industrial waste and by-products as biostimulants in plants plays a role in a series of changes affecting photosynthesis, antioxidant enzyme activity, plant metabolite activities, and molecular and physiological processes (Zhang et al., 2024).

Medicinal and aromatic plants are the greatest sources of phenolic compounds that exhibit biostimulant properties (Kisiriko et al., 2021). These bioactive compounds can be identified and characterized from various plant parts such as leaves, stems, flowers and fruits (Azmir et al., 2013). In the experiment, high levels of 1.8-cineole were determined in the essential oil content of rosemary leaves (Table 1). 1.8-cineole is a natural monoterpene, also known as eucalyptol. It is known as the main component of the rosemary plant. α-Pinene is an organic compound from the terpene class. Commercially important derivatives of α-pinene are linalool, geraniol, nerol, α-terpineol and camphene (Sell, 2000). In the study, 8.76% α-pinene, 2.31% linalool, and 8.31% α-terpineol were determined in rosemary essential oil (Table 1). These substances are important compounds that play a role in the antioxidant defence systems of plants (Suffredini et al., 2004). Volatile oil contents may vary depending on cultivation and harvest conditions. This supports that the observed positive effect of by-product treatments may be variable. Another issue is the possibility of allelopathic effects of the active ingredients in the volatile oil composition. This situation has revealed the importance of dose optimization in seed priming studies (Oğuz, Oğuz & Güler, 2023). It is also recommended to elucidate the long-term effect of by-product applications on the soil biome.

In the by-product hormone analysis, both ABA and GA3 contents were at similar levels. IAA contents were found to be higher in PE than in PB. Phytohormones such as auxins, gibberellins and cytokinins found in plant extracts and biomass are growth promoters that can be evaluated differently in agricultural systems. Moreover, a strong correlation was pointed out with the phenolic compounds and secondary metabolites they contain as the main contributors to the biostimulant, bioprotective and antioxidant capacities of plant extracts and essential oils (Martins et al., 2011). The study supports the idea that the bioregulatory effects of PE and PB by-products are closely related to the contained compounds.

Researchers have reported that phenolic compounds in plant extracts and essential oils have biostimulant and antioxidant effects on seed germination, rooting and shoot development by treating to seeds, leaves or soil (Kisiriko et al., 2021; Oğuz, Oğuz & Güler, 2023). Biostimulants cannot provide nutrients alone, unlike fertilisers (Van Oosten et al., 2017). Biostimulants obtained from natural sources provide benefits by stimulating the mechanisms involved in biochemical events and physiological processes (Madende & Hayes, 2020). This effect primarily aims to protect the tissue from oxidative stress by preventing the breakdown of auxin in the plant. Balancing hormone activity and suppressing oxidative stress-related processes are of great importance in early physiological development such as germination, root development and plant height (De Klerk et al., 2011). The significant positive changes in plant height and chlorophyll content of PE and PB treatments compared to control T0, especially in the early stages of growth (seedling and tillering), are thought to be hightly related to the ABA, GA3 and IAA content in PE and PB (Fig. 3).

The stimulatory activity of plant extracts, pulps, essential oils and phenolic compounds has been reported to show an increase in the amounts of beneficial metabolites that play a role in growth and development when applied to seeds or seedlings (Ben-Jabeur et al., 2022; Oğuz, Oğuz & Güler, 2023). The main metabolites induced in the plant by exogenous treatments of plant extracts include chlorophyll, carotenoids, proteins, sugars and amino acids (Kisiriko et al., 2021). Chlorophyll is one of the main chloroplast components for photosynthesis, and chlorophyll content has a positive relationship with the rate of photosynthesis (Rama Reddy et al., 2014). From the early stages of development, PE and PB treatments alone or together caused an increase in chlorophyll content compared to the control.

Increased amounts of proline under stress conditions play an important role in the activation of stress mechanisms associated with reactive oxygen species (ROS) (Rao & Chaitanya, 2016). Farooq et al. (2018) reported that exogenous treatment of plant extracts improved the accumulation of proline, glycine betaine and other soluble phenolics, thus increasing tolerance to biotic and abiotic stresses. Another substance produced by membrane lipids in response to ROS is malondialdehyde (MDA). MDA can be used as an indicator to evaluate the degree of plasma membrane damage at the cellular level and the response of plants to stress. Similar to proline accumulation, T1(PE) and T3 (PE+PB) treatments in the early growth period (seedling and tillering) caused an increase in MDA content (Fig. 3). It can be interpreted that the seed priming process activated stress mechanisms in the early development period in both treatments (T1 and T3). Ibrahim (2016) reported that seed priming can create a “stress memory” in germinating seeds by creating stress conditions that induce enzyme activation and osmotic adjustment in the seed. In the study results, the increase in stress-related biochemicals during the early development period with seed priming treatment supports the idea that the priming process induces effective mechanisms in stress tolerance.

RWC is one of the important parameters used to predict stress tolerance and plant growth (Ahmad et al., 2022). Stress tolerance is known to be associated with high water potential in tissues. The water potential of plants under stress conditions is lower than that of plants under non-stress conditions (Eastham, Oosterius & Walker, 1984). Oğuz, Oğuz & Güler (2023) reported a significant relationship between proline content and RWC. According to the results, the contents of chlorophyll, proline, MDA, and RWC varied according to developmental stages. It is known that the responses of plants to environmental factors vary according to their developmental periods (Oguz et al., 2022). RWC represents water availability within the plant as a result of water loss through transpiration and uptake by the roots (Georgii et al., 2017). Leaf water potential, which is important for plant survival and photosynthetic processes, is closely related to turgor pressure, stomatal closure and cell growth (Alghabari et al., 2015). RWC protection provides tolerance to low and moderate water stress. On the other hand, the decrease in photosynthetic efficiency has been directly associated with the decrease in RWC (Nikinmaa et al., 2013). According to the study results, a link can be established between the high chlorophyll content from the T3 treatment and the increases in RWC content. Although the accumulation of stress-related biochemicals is effective, the increase in the amount of chlorophyll has been accepted to have a positive effect on the water cycle within the plant by triggering physiological processes.

Development is the progression of the plant life cycle. Development is divided into two groups: vegetative and generative within this cycle (Oguz et al., 2022). Both the onset and duration of these critical developmental stages are affected by environmental stresses. Early seedling and tiller development are important because the number of tillers per plant is a critical yield component (Klepper, Rickman & Peterson, 1982). Besides, early maturing genotypes are preferred due to their ability to escape drought, heat stress, disease, pests and other stresses at the end of the growing season (Zikhali et al., 2014). Shortening the production period and rapid completion of developmental stages through treatments play an important role in combating environmental stresses. In the field experiment, significant shortenings were obtained in the development processes of wheat seedlings, such as emergence, TD, SED, and HD with by-product treatments. By-product treatment has significant potential in the rapid development of wheat in adverse climate and environmental conditions.

Producers can optimize tillering numbers and ultimately contribute to higher yields by ensuring proper management before planting and until early development (Alley et al., 2009; Tilley, Heiniger & Crozier, 2019). In this process, a single plant can produce up to five tillers (although environmental conditions, cultivar selection and practices are influential). Tillering is directly related to yield. It has been reported that tillers produced before January contribute approximately 87% of the grain yield under field conditions, while tillers produced in January, February and March can only contribute 2–11% of the yield (Oakes, 2023). Therefore, in the field experiment, tiller numbers were measured in the last week of December. Accordingly, single and combined application of by-product treatments increased tillering numbers (Fig. 5).

Figure 5 Expected social impact of the study.

Illustration abstract.

Plants experience significant changes in the flowering stage after the vegetative stage ends. A key point in the wheat life cycle is the transition of the shoot tip from the vegetative to the generative development stage (Waddington, Cartwright & Wall, 1983). The production of new leaf primordia ceases and spikelet formation begins. This represents a determining the final leaf number (Wang et al., 1995). During this period, the flag leaf alone provides approximately 60% of the daily photosynthetic activity (Racz et al., 2022). On the other hand, flag leaf has an important role in determining the grain filling rate and final yield (Liu et al., 2021). In greenhouse and field experiments, it was determined that T2 and T3 treatments generally showed significant positive effects compared to the T0 control treatment in the parameters examined related to yield. The results support the idea that the final yield can be increased by improving the parameters affecting the yield by the by-product exogenous treatment. In the variety studies conducted by the Central Research Institute of Field Crops of the Republic of Turkey, an average yield of 194.8 kg/da was determined in the agricultural value measurement trials of the variety candidate (ANK-KA02/20) belonging to the Kavılca population (Variety Registration and Seed Certification Center of Turkey, 2023). In the study, the highest yield determined in the T3 treatment (264.96 kg da−1) is higher than a typical Kavılca yield. However, these values are possible to vary depending on climate, environment and growing conditions. According to the meta-analysis results of Mickky (2022), different seed priming techniques caused a significant increase in the economic yield (29%), biological yield (22%) and thousand-grain yield (16%) of wheat. It was also stated that the studies showed high consistency with each other. Extracts obtained from plant parts allow for cheap and easily repeatable applications compared to chemical inputs.

During maturation, nutrient assimilation gives way to nutrient remobilization. Increasing amounts of nutrients (amino acids, simple sugars, mineral nutrients) are transported into the grain (Feller & Fischer, 1994). It has been stated that 80–89% of the nitrogen in the grain is transported by remobilization, although it varies according to genotypes (Kichey et al., 2007). Amino acids transported from the stem and leaves to the grain during senescence are used in the synthesis of grain proteins in the embryo, aleurone and endosperm (Barneix, 2007). Classification of grains according to protein content as 9% and below is very low, 11.6–13.5% is medium, and 13.5–15.5% is high protein (Williams et al., 1988). The effect of the T3 treatment on GPC and GNC values increased significantly compared to other treatments in the greenhouse experiment (Table 3). In the field experiment, the most effective treatment on grain nitrogen content was T3; however, no significant positive effect was observed on grain protein values (Table 6).

Environmental stress factors can affect the vegetative and generative stages of the plant differently (Oguz et al., 2022). The ultimate goal served by all developmental stages is yield. Physiological adaptations such as efficient use of water in the vegetative process, low stomatal movement and maintenance of balance in turgor pressure ensure the conservation of water needed by the plant until the grain-filling stage in the generative period (Lopez et al., 2017). The mechanisms physiologically triggered by seed priming have an effect in the later stages of growth (Oğuz, Oğuz & Güler, 2023). On the other hand, by-product (PB) treatment is thought to continue to support the production process by providing long-term biostimulant presence in the soil. Since all parameters examined during the development and growth process are interrelated, physiological and biochemical processes supported from the early stages of growth ultimately affect the yield. This study revealed successful results that will contribute to the importance of evaluating plant by-products as biostimulants in increasing yield performance in rainfed wheat agriculture.

Economy, society and environment are interrelated components that together represent sustainability as a common denominator (Purvis, Mao & Robinson, 2019). Negativity in any of these will affect the final result. Each of the sustainable development goals forms part of an ultimate goal. Therefore, sustainability represents a social goal as a whole. Sustainability will reach the goal with economic and social development on an environmental basis. The expected social impact of our study is that environmentally friendly and renewable applications with by-products are likely to increase production and quality in agriculture, alleviate productivity losses caused by climate change and environmental stress factors, enable farmers to develop economically and health-wise and improve social welfare (Fig. 5).

Multiple factors interact with field conditions and cannot be controlled. Important findings to be obtained under controlled conditions should therefore be tested in field conditions. The important thing is that the research results can reach from the scientific level to the production level. Integration of trials into production systems and evaluation under farmer conditions will trigger widespread social interaction. This study aims to balance agricultural inputs and yield as sustainable and economical. The results can make a significant contribution to increasing the social and economic of smallholder farmers.

The waste left over from essential oil production is not used. The fact that these by-products can be reused without the need for reprocessing is cost-free for farmers. In addition, the ease of seed priming and the ability to mix plant biomass into the soil with planting provide ease of application in a single stage (single period) without the need for repetition. In addition, it does not require tools, infrastructure and labor for application. Undoubtedly, the increase in yield with minimum input will be effective in increasing profitability. In addition, the reuse of waste can provide a large-scale economic added value.

Conclusions

Elimination/minimization of chemical inputs in agriculture is of great importance in achieving the “Sustainable Development Goals”. When the increasing costs of chemical inputs are taken into account, farmers will prefer alternative sustainable methods that are cheap and environmentally friendly. Especially in rain-based production systems where chemical inputs are minimized, such simple but effective treatments are successful in mitigating yield losses caused by climate change. The treatments of plant-based seed priming methods developed in parallel with the same purpose method in low concentrations can be made sustainable for smallholder farmers as an advantage. Sustainable effective, simple, and low-cost practices and maintaining the economic income levels of smallholder farmers will be effective in alleviating forced migration from rural areas to cities. In this context, it is recommended that efforts to prevent yield losses in agricultural production should focus on the social, economic and cultural quality of life of producers and their impact on social development from a wide perspective. Moreover, this study includes the results of rosemary by-product trials conducted on a Kavılca wheat. It is recommended to test different target plant species and other by-products concentration/solution to determine the widespread effect.

Supplemental Information

Supplemental Information 1 Temperature and precipitation data of the field experiment area for 2022–2024

Supplemental Information 2 Raw data of field experiments

Supplemental Information 3 Raw data of greenhouse experiments

Additional Information and Declarations

Competing Interests

Author Contributions

Data Availability

The authors declare there are no competing interests.

Muhammet Cagri Oguz conceived and designed the experiments, performed the experiments, analyzed the data, prepared figures and/or tables, authored or reviewed drafts of the article, and approved the final draft.

The following information was supplied regarding data availability:

The greenhouse and field raw data are available in the Supplemental Files.

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
