# Peer review of "A sustainable approach for smallholder farmers: evaluation of plant by-products and industrial waste in wheat cultivation"

_PeerJ, doi:10.7717/peerj.19775_

## Round 0.1 · original submission · Major Revisions

**Language Note:** The review process has identified that the English language must be improved. PeerJ can provide language editing services - please contact us at [email protected] for pricing (be sure to provide your manuscript number and title). Alternatively, you should make your own arrangements to improve the language quality and provide details in your response letter. – PeerJ Staff

Reviewer 1 ·

Basic reporting

The author has written the manuscript in professional English that enables audiences from multiple nations to understand its content easily. The first section presents enough information about smallholder agriculture and rainfed systems and plant by-products as biostimulant possibilities by citing appropriate literature sources (FAO, 2012; Lowder et al., 2016; Ricciardi et al., 2018). The paper follows PeerJ’s structural template through its sections for Abstract, Introduction and Materials and Methods, Results, Discussion followed by Conclusions. The article contains appropriate figures and tables which properly support the established hypotheses but requires additional enhancements (as mentioned below).
Suggestions for Improvement:
1. Minor language issues in both language and grammar are present which would enhance the document's clarity. For example:
Line 110: "anayles" should be "analyses."
The spelling of "Kjchdahl" represents an incorrect term which should read as "Kjeldahl" on Line 185.
2. Figures and Tables:
The hormone contents data in Figure 2 exists only as textual information without any supporting graphical element. The addition of a visual representation in this format would help readers understand the information better.
Figure 3 (biochemical analyses) is referenced but not provided in the document. Ensure all figures are included and clearly labeled with statistical significance indicators (e.g., letters for Duncan test results).
Table 1 (GC-MS results) is clear, but consider adding retention times or a reference to the method for reproducibility.

Experimental design

The study is original, within PeerJ’s scope, and addresses a relevant question on using hydrodistillation by-products as bioregulators in wheat agriculture. Greenhouse and field experiments use a rigorous randomized design, meeting high technical and ethical standards. Methods are mostly detailed enough for replication, covering hormone analysis, GC-MS, and biochemical measurements.
Suggestions for Improvement:
1. Clarify Methods:
Specify P_E concentration for seed priming (Line 103).
Describe field soil characteristics and P_B application method (Line 161).
Confirm statistical assumptions (normality, variance homogeneity) for ANOVA (Line 193).
2. Control Conditions: Clarify if control (T_0) received water priming to match P_E treatment.
3. Ethics: Explicitly state no synthetic inputs were used (Line 167) to emphasize sustainability.

Validity of the findings

Raw data are provided, statistically sound (ANOVA, Duncan’s test), and support the conclusion that by-products enhance wheat yield. Greenhouse and field results are consistent, and the discussion aligns findings with literature (e.g., Mickky, 2022).
Suggestions for Improvement:
1. Data Interpretation:
Link hormone levels (ABA, GA3, IAA) to physiological effects in Discussion (Line 206).
Support or temper “stress memory” claim for proline/MDA with evidence (Line 232).
2. Statistical Reporting: Include p-values or test statistics in tables (Tables 2-5).
3. Contextualization: Compare T_3 yield (264.96 kg da⁻¹) to typical Kavilca wheat yields (Line 339).
4. Limitations: Note single wheat variety and rosemary by-products; suggest testing other varieties/species

Additional comments

The manuscript is a valuable contribution to sustainable agriculture, particularly for smallholder farmers in rainfed systems. The use of rosemary by-products as biostimulants is innovative and aligns with global sustainability goals. The authors are commended for their comprehensive dataset, clear presentation of results, and focus on practical applications. However, the following points could further strengthen the manuscript:
1. Introduction (Lines 28-81): The introduction is thorough but lengthy. Condense the discussion of smallholder farming (Lines 29-36) to focus more on the specific problem of waste utilization and its relevance to wheat.
2. Discussion (Lines 343-531): The discussion is well-supported but repetitive in places (e.g., Lines 419-430 repeat points about proline and MDA). Streamline to avoid redundancy and focus on novel insights.
3. Social Impact (Lines 532-557): Figure 5 is a strong visual, but the accompanying text is overly general. Provide specific examples of how the study’s findings could be implemented by farmers (e.g., cost estimates for by-product processing).
4. References: The reference list is extensive, but some citations (e.g., Ritchie & Roser, 2024) use a future date. Correct these to reflect the actual publication date (e.g., 2021, as per the manuscript’s context).

Reviewer 2 ·

Basic reporting

The areas which need to improve,
1) literature references, some of them very old
2) discussion, its repetition of results
3) language

Experimental design

the information of green house experiment is missing which need to add.

Validity of the findings

no comment

Additional comments

The manuscript is generally well-organized and addresses a relevant and timely topic in sustainable agriculture. The background provides a clear foundation for using plant-based biostimulants, especially in the context of smallholder farming and rainfed agriculture. Here are some comments that help the author improve the manuscript's structure.
Line 41: Can the author provide the latest reference with an updated number, “approximately 60% of farmers earn income by producing with the rainfed-based production model”?
Line 42: Add the latest study.
HPLC and GC-MS Methodology: Some details are missing from the analytical methods—e.g., column temperature programming, run times, and standard calibration curves.
Line 132-134: Please add more details of the greenhouse experiment, how it was conducted, including the details about environmental conditions.
Revise the statistical analysis section on how the author performs all the analysis, such as which ANOVA was used, One-way?
In the Discussion section, the author repeated the findings that were already stated in the Results. Improve it by discussing the results and supporting it with relevant studies.
There is minimal discussion of potential limitations (e.g., variability in by-product composition, long-term soil impacts). Please include a short section acknowledging limitations and potential confounders.
Some references are very old in the discussion, it is suggested that the author replace them with the latest references.
It is suggested that the author add future perspectives and practical applications of this study in the conclusion section.
The quality of figures 2 and 3 is low, which needs to improve, and some letters (a, b…) are missing.
Please use the species name in italics in the reference list and ensure consistency throughout the manuscript.

Annotated reviews are not available for download in order to protect the identity of reviewers who chose to remain anonymous.

---

## Round 0.2 · accepted · Accept

Based on the reviewers’ comments, both of whom accepted the author's rebuttal and recommended acceptance, I confirm that the manuscript is hereby accepted for publication.

Reviewer 1 ·

Basic reporting

The English used is clear and professional.

The introduction gives a good background.

References are relevant, but a few recent ones (2021–2024) can be added.

The structure follows journal standards.

Figures and tables are good.

Experimental design

The study is original and fits the journal's scope.

The objective is relevant.

Standard methods are used, but details like sample size and number of repeats should be mentioned clearly.

No ethical concerns.

Validity of the findings

The tests and results are valid, but it’s unclear how many replicates were done.

Raw data behind graphs should be included as supplemental material.

Conclusions match the results, but the practical use in product development could be explained further.

Additional comments

Add 1–2 newer references to show updated work.

Reviewer 2 ·

Basic reporting

All the comments are addressed well by the author.

Experimental design

All the comments are addressed well by the author.

Validity of the findings

All the comments are addressed well by the author.